# Mitochondrial Dysfunction in Glycogen Storage Disorders (GSDs)

**DOI:** 10.3390/biom14091096

**Published:** 2024-09-01

**Authors:** Kumudesh Mishra, Or Kakhlon

**Affiliations:** 1Department of Neurology, The Agnes Ginges Center for Human Neurogenetics, Hadassah-Hebrew University Medical Center, Jerusalem 9112001, Israel; 2Faculty of Medicine, Hebrew University of Jerusalem, Ein Kerem, Jerusalem 9112102, Israel

**Keywords:** mitochondrial dysfunction, glycogen storage disorders, reactive oxygen species, oxidative stress, autophagy and mitophagy, myopathy

## Abstract

Glycogen storage disorders (GSDs) are a group of inherited metabolic disorders characterized by defects in enzymes involved in glycogen metabolism. Deficiencies in enzymes responsible for glycogen breakdown and synthesis can impair mitochondrial function. For instance, in GSD type II (Pompe disease), acid alpha-glucosidase deficiency leads to lysosomal glycogen accumulation, which secondarily impacts mitochondrial function through dysfunctional mitophagy, which disrupts mitochondrial quality control, generating oxidative stress. In GSD type III (Cori disease), the lack of the debranching enzyme causes glycogen accumulation and affects mitochondrial dynamics and biogenesis by disrupting the integrity of muscle fibers. Malfunctional glycogen metabolism can disrupt various cascades, thus causing mitochondrial and cell metabolic dysfunction through various mechanisms. These dysfunctions include altered mitochondrial morphology, impaired oxidative phosphorylation, increased production of reactive oxygen species (ROS), and defective mitophagy. The oxidative burden typical of GSDs compromises mitochondrial integrity and exacerbates the metabolic derangements observed in GSDs. The intertwining of mitochondrial dysfunction and GSDs underscores the complexity of these disorders and has significant clinical implications. GSD patients often present with multisystem manifestations, including hepatomegaly, hypoglycemia, and muscle weakness, which can be exacerbated by mitochondrial impairment. Moreover, mitochondrial dysfunction may contribute to the progression of GSD-related complications, such as cardiomyopathy and neurocognitive deficits. Targeting mitochondrial dysfunction thus represents a promising therapeutic avenue in GSDs. Potential strategies include antioxidants to mitigate oxidative stress, compounds that enhance mitochondrial biogenesis, and gene therapy to correct the underlying mitochondrial enzyme deficiencies. Mitochondrial dysfunction plays a critical role in the pathophysiology of GSDs. Recognizing and addressing this aspect can lead to more comprehensive and effective treatments, improving the quality of life of GSD patients. This review aims to elaborate on the intricate relationship between mitochondrial dysfunction and various types of GSDs. The review presents challenges and treatment options for several GSDs.

## 1. Background

Glycogen storage disorders (GSDs) are a group of inherited glycogen metabolic disorders caused by deficiency of enzymes involved in the synthesis or degradation of glycogen. Most GSDs are congenital and can present at any age, from newborn to adulthood [1]. GSDs primarily affect the liver and skeletal muscles, where glycogen is predominantly stored; however, depending on the specific enzyme deficiency and its distribution across various tissues, other organs, including the heart, kidneys, and brain, can also be affected [2,3,4]. GSDs are classified according to the specific enzyme deficiency and the affected organ and include GSD 0a, GSD 0b, GSD I, GSD II (Pompe disease, PD), GSD III, GSD IV, GSD V, GSD VI, GSD VII, GSD IX, GSD X, GSD XI (lactate dehydrogenase subunit A deficiency), GSD XII, GSD XIII, phosphogluco-mutase 1 deficiency–congenital disorder of glycosylation (PGM1-CDG; previously called GSD XIV), GSD XV, Fanconi–Bickel syndrome (FBS), and phosphoglycerate kinase (PGK) deficiency [1,4]. Mitochondria are essential organelles known for their role in ATP production through oxidative phosphorylation (OxPhos), regulation of cellular metabolism, and involvement in apoptosis. Impairment in mitochondrial function can have widespread effects on cellular energy status and metabolic homeostasis [2,5,6]. In GSDs, mitochondrial dysfunction can exacerbate the metabolic derangements caused by impaired glycogen metabolism, leading to more complex pathological conditions and influencing disease outcomes [7,8,9]. Due to remarkable changes in glucose and fuel supply, characterizing most GSDs, mitochondrial dysfunction is assumed to be a critical component in the pathology of various GSDs. The abnormal accumulation or deficient production of glycogen, primarily in the liver and muscles, results in a spectrum of clinical manifestations. Although traditionally, the focus has been on the direct consequences of enzyme deficiencies on glycogen metabolism, recent studies have shed light on the significant role of mitochondrial dysfunction in the progression and severity of GSDs. One of the most extensively studied GSDs with respect to mitochondrial dysfunction is GSD type II (PD) [10]. Pompe disease results from mutations in the acid alpha-glucosidase (GAA) gene, leading to the accumulation of glycogen within lysosomes. This lysosomal storage disorder primarily affects muscle cells, including cardiac and skeletal muscles. Recent research has indicated that secondary mitochondrial dysfunction plays a pivotal role in the disease’s pathophysiology. The accumulated glycogen disrupts normal cellular processes, leading to mitochondrial swelling, reduced mitochondrial membrane potential, and impaired OxPhos. These mitochondrial abnormalities contribute to muscle weakness and respiratory complications, which are characteristic of PD. Similarly, GSD III (Cori disease) is caused by a deficiency in the glycogen debranching enzyme (amylo-1,6-glucosidase, 4-alpha-glucanotransferase) accumulating abnormal glycogen with shorter outer branches. This condition not only affects liver and muscle tissues but also shows a significant impact on mitochondrial function. Studies have demonstrated that patients with GSD III exhibit reduced activities of mitochondrial respiratory chain complexes, increased oxidative stress, and altered mitochondrial morphology. These mitochondrial impairments are believed to contribute to the muscle weakness, cardiomyopathy, and hepatic dysfunction observed in GSD III patients [11,12,13]. The interplay between glycogen storage and mitochondrial dysfunction is also evident in GSD V (McArdle disease), which is caused by a deficiency in myophosphorylase, an enzyme crucial for glycogen breakdown in muscle cells [14]. The inability to mobilize glycogen during exercise leads to muscle cramps, fatigue, and, in severe cases, rhabdomyolysis. Recent studies have highlighted that mitochondrial dysfunction, characterized by decreased mitochondrial biogenesis and impaired oxidative metabolism, exacerbates the exercise intolerance observed in McArdle disease. Mitochondrial dysfunction has emerged as a critical component in the pathophysiology of several GSDs, influencing the clinical manifestations and therapeutic approaches.

Based on an extensive literature search using keywords like mitochondrial dysfunction in GSDs, mitochondrial impairment and GSDs, metabolic disorders and GSDs, mitochondria-related myopathy, and autophagy in GSDs, we went through the literature and followed major abnormalities related to mitochondrial dysfunction across different GSDs. This review provides an extensive overview of mitochondrial dysfunction in GSDs and discusses the underlying mechanisms, clinical implications, and potential therapeutic strategies.

## 2. Glycogen Storage Disorders (GSDs) an Overview 

GSDs primarily affect the liver and skeletal muscles due to the high abundance of glycogen in these tissues. Glucose and glycogen convert into one another via synthesis or degradation through various steps in the glycogen metabolism pathways chiefly in the liver and skeletal muscles, catalyzed by different metabolic enzymes. Mutations in genes encoding individual enzymes in the glycogen metabolism pathway led to GSDs, depending on the enzyme defect and its relative expression in the liver and skeletal muscle, as well as in the kidney, heart, and brain. Also, clinical manifestations of GSDs vary from one disorder to the other. Poor glycogen catabolism to glucose in GSDs can lead to Fasting hypoglycemia with ensuing hepatomegaly. Muscle GSDs are present in one of two different ways: exercise intolerance with rhabdomyolysis or fixed muscle weakness without rhabdomyolysis. GSDs are categorized based on the specific enzyme deficiency and the resultant metabolic abnormalities; details are summarized in (Table 1) and illustrated in Figure 1. 

## 3. Mitochondrial Metabolic Dysfunction

Mitochondria are essential organelles responsible for energy production through OxPhos, regulation of apoptosis, and maintenance of cellular metabolism. Mitochondria also play a role in calcium homeostasis, reactive oxygen species (ROS) production, and biosynthesis of certain macromolecules. Most of the cell’s oxidative metabolism takes place in mitochondria, and electron leak in OxPhos is considered the major producer of ROS in the cell. Mitochondria are also the major sites of energy production, of generation of reducing equivalents (NADH and FADH_2_) in the tricarboxylic (TCA) cycle, and of fatty acid β-oxidation [50]. Primary mitochondrial impairment leads to cardiomyopathy and progressive, muscular, and neuronal degeneration. Defective OxPhos can lead to an overproduction of ROS that damages DNA, proteins, and lipid membranes by oxidative reactions [51]. Lactate dehydrogenase D (LDHD), whose deficiency is sometimes referred to as GSD XI, catalyzes the conversion of pyruvate to lactate as a terminal step in glycolysis. OxPhos inhibition decreases the NAD^+^/NADH ratio and thus increases LDH activity, which oxidizes NADH to NAD^+^ while reducing pyruvic acid to lactate. Deficient activity of OxPhos complexes can cause severe pathological manifestations from neonatal to adult ages, including fatal infantile lactic acidosis possibly due to glycolytic overcompensation, infancy/early childhood onset of neuropathological disorders [52], cardiomyopathy, liver disease, and myopathy [53]. Mitochondrial impairment was found in patients with GSD I, GSD III, GSD VI, and GSD IX by examining enzymatic activities in lymphocytes. The results revealed significant alterations in enzyme activity profiles across all GSD types. Notably, reduced activity of succinate dehydrogenase (a component of complex II of the respiratory chain) and glycerol-3-phosphate dehydrogenase (a glycolytic enzyme) was observed. 

Conversely, decreased enzymatic activity of NADH dehydrogenase and lactate dehydrogenase (LDH) was observed in all GSD types, with the most pronounced changes found in GSD I [54]. These findings provide evidence of mitochondrial dysfunction in various forms of GSDs, suggested by the decreased NAD^+^/NADH ratio, which is consistent with the inhibition of NADH oxidation in the mitochondrial respiratory chain. Figure 2 highlights the complex interplay between metabolic pathways and mitochondrial function in these disorders. In GSDs, the normal metabolism of glycogen is disrupted, leading to the impaired breakdown of glycogen and ensuing insufficiency of its degradation product glucose. Since glucose serves as a major OxPhos fuel, researchers have investigated the link between deficient glycogen metabolism and OxPhos insufficiency in GSD pathogenesis. A reduction in the use of carbohydrates as fuel and reduced regeneration of phosphorylated molecules in mitochondria during exercise has been reported [55,56,57]. GSD Ia patient fibroblast exhibited a lower concentration of malate, responsible for translocating electrons produced during glycolysis across the semipermeable inner membrane of the mitochondrion for OxPhosS [58]. Peripheral lymphocytes from GSD I patients have shown decreased activity of succinate dehydrogenase and NADH dehydrogenase [54]. A previous study has shown the presence of indirect markers of TCA cycle and fatty acid oxidation (FAO) overload in the urine and plasma of GSD Ia patients [59]. Accumulation of free carnitine and long-chain acylcarnitines, a transporter of fatty acids, into the mitochondria for subsequent β-oxidation, has been observed, and also increased secretion of various TCA cycle metabolites was observed in the urine of GSD I patients [60,61]. The cause of mitochondrial dysfunction in GSDs is not fully known. A decrease in mitochondrial oxidation could result solely from reduced mitochondrial content. In the GSD Ia animal model of G6pc^−/−^ mice, several mitochondrial dysfunctions have been identified in the liver, including reduced mitochondrial content, abnormal morphology, impaired respiration, and disturbed TCA cycle function. Significant abnormalities in liver mitochondria were observed, including reduced mitochondrial content, irregular ultrastructure of cristae, and overall morphological alterations. Additionally, there are indications of disrupted mitophagy, the process by which cells remove damaged or dysfunctional mitochondria. Ultrastructural analyses of knockout (KO) mice and knockdown (KD) AML-12 cells show a significant number of damaged mitochondria. The changes in mitochondrial membrane potential might also cause defects in membrane structure, possibly due to altered or imbalanced incorporation of fatty acids into the mitochondrial inner membrane [12]. Glucotoxicity or lipotoxicity due to lipid species such as saturated free fatty acids and ceramides might also mediate mitochondrial damage. It is possible that excessive intracellular carbohydrates or lipids could lead to mitochondrial dysfunction [62,63]. Mitochondrial dynamics, including fusion and fission, are crucial for maintaining mitochondrial function and integrity [64], and when mitochondrial fission is compromised, compensatory glucose uptake and glycogen synthesis are activated [65]. In several GSDs, abnormal glycogen accumulation disrupts these processes. In GSD III, the accumulation of structurally abnormal glycogen affects mitochondrial morphology, leading to impaired fusion and fission. This results in dysfunctional mitochondria that are less efficient in energy production and more prone to oxidative damage [12]. Progressive skeletal muscle hypotonia and cardiac hypertrophy are classical manifestations of infantile-onset PD [66]. PD patients present with hypertrophic cardiomyopathy, disrupted metabolic signaling, and mitochondrial dysfunction [67]. In damaged mitochondria observed in PD, ATP synthase activity and ensuing ATP production are impaired by depolarization of the inner mitochondrial membrane potential, resulting in energetic deficit and contractile failure in muscle tissues [68]. Huang et al. carried out a study in PD-specific induced pluripotent stem cells (PD-iCMs), which exhibited a significant reduction in OxPhos compared to control-iCMs. This decreased OxPhos was partially ameliorated by treatment with recombinant alpha-glucosidase (rhGAA), which also led to increased basal respiration, maximal respiration, and spare respiratory capacity, along with a subsequent rise in proton leakage and ATP production. PD-iCMs had a depolarized mitochondrial membrane potential and elevated reactive oxygen species (ROS) production relative to Ctrl-iCMs, both of which were mitigated by rhGAA treatment. These findings suggest that, in addition to reducing lysosomal glycogen accumulation, rhGAA can partially restore mitochondrial function, possibly by GAA-dependent improved lysosomal function and resultant improved mitophagy which might enhance mitochondrial quality control. These improvements might potentially tackle the cardiac pathology observed in PD [8]. Another study in PD-iCMs reported markedly swollen cristae and mitochondrial dysfunction, including decreased glycolysis and OxPhos in [69]. Lim et al. identified mitochondrial dysfunction associated with abnormal energy metabolism in skeletal muscle biopsies of patients and in animal models of PD [70]. 

### 3.1. Overproduction of ROS and Oxidative Stress 

In addition to ATP production, mitochondria are also a key source of reactive oxygen species (ROS) [71]. ATP is generated by the electron transport chain (ETC), where low-energy, high-redox potential electrons are transferred to oxygen in a controlled manner. Occasionally, some electrons deviate from this path and directly react with oxygen, forming superoxide radicals. These superoxide radicals can then be converted, either enzymatically by superoxide dismutase (SOD) or spontaneously, into hydrogen peroxide [72]. Hydrogen peroxide can participate in the Fenton reaction to produce hydroxyl radicals, which are highly reactive and capable of causing significant damage to cellular structures such as membranes, proteins, enzymes, and nucleic acids, potentially leading to cell death [73,74]. To mitigate ROS damage, mitochondria possess a sophisticated antioxidant defense system. SOD converts superoxide radicals into hydrogen peroxide, which is then reduced to water by glutathione peroxidase in the presence of reduced glutathione (GSH), which is oxidized during this reaction to glutathione disulfide (GSSG). This process effectively reduces the formation of harmful hydroxyl radicals, thereby scavenging most ROS generated within the mitochondria (which, in general, is estimated to be more than 90% of the ROS produced in cells [75]. Additionally, hosting antioxidant enzymes, such as SOD, glutathione peroxidase, glutathione reductase, and catalase, and antioxidant molecules, such as coenzyme Q, mitochondria assist in scavenging ROS originating from other cellular organelles, such as peroxisomes [76]. When ROS production exceeds the capacity of the cellular antioxidant defenses, oxidative stress ensues, leading to damage to cellular macromolecules and impairment of cellular functions and viability [77]. Oxidative stress is increasingly recognized as a pivotal mechanism in the development of various diseases, including metabolic syndrome (MetS). Increased oxidative stress exacerbates mitochondrial dysfunction by damaging components of the electron transport chain (ETC) and other mitochondrial constituents; it enhances mitochondrial fragmentation and furthers the decline in OxPhos [78] (Figure 3). Excessive ROS production can damage key mitochondrial components such as NADH dehydrogenase, cytochrome c oxidase, and ATP synthase, leading to a shutdown of mitochondrial energy production. ROS production is also heavily involved in the resulation of intracellular calcium. Under normal physiological conditions, calcium (Ca^2+^) fluxes are tightly regulated across the plasma membrane and between intracellular compartments. However, excessive ROS generation can directly damage Ca^2+^ regulating proteins, and voltage-gated Ca^2+^ channels in the plasma membrane, Ca^2+^ ATPases in the endoplasmic reticulum, and mitochondrial electron transport chain proteins. This results in elevated intracellular Ca^2+^ levels, disturbing Ca^2+^ homeostasis and further contributing to mitochondrial damage, disruption of the mitochondrial respiratory chain, and increased mitochondrial membrane permeability. Since GSDs also usually implicate mitochondrial damage due to fuel restriction and physical damage to mitochondria by accumulated glycogen (e.g., in PD), these mitochondrial dysfunctions can be implicated in GSD disease pathology [23,79]. 

### 3.2. Mitochondrial Biogenesis 

Mitochondria contain their own DNA, which encodes a few components of the ETC and 22 mitochondrial tRNAs [80]. The majority of ETC components and other mitochondrial machinery are derived from nuclear genes. Therefore, mitochondrial biogenesis requires coordinated expression of both nuclear and mitochondrial genes [80,81]. Peroxisome proliferator-activated receptor gamma coactivator-1 alpha (PGC-1α) is a crucial regulator of this process, activating various transcription factors like Nuclear respiratory factors (NRF-1 and NRF-2) and estrogen-related receptors (ERRs), which induce mitochondrial transcription factor A (TFAM), to promote mitochondrial gene transcription [82,83]. PGC-1α also enhances mitochondrial fatty acid oxidation by co-activating PPARα and PPARδ, leading to increased mitochondrial mass and substrate oxidation [84]. PGC-1α activation, triggered by elevated AMP (via AMPK) and increased NAD+ (via Sirtuin-1) (Figure 4), also reduces cellular oxidative stress by boosting mitochondrial antioxidant enzyme expression [85,86]. Thus, PGC-1α is a key target for therapeutic strategies against metabolic syndrome (MetS), presumably obtained by enhancing mitochondrial biogenesis. Mitochondrial biogenesis increases mitochondrial turnover, replacing damaged, pro-oxidative mitochondria with new mitochondria, which produce relatively fewer ROS, thus improving mitochondrial efficiency and reducing oxidative stress [86,87]. Assuming that increased ROS production might contribute to the pathogenesis of MetS through mechanisms involving oxidative stress [79], the anti-oxidative mitochondrial biogenesis could counter MetS. In a study using L-G6pc−/− mice, liver-targeted G6Pase-α deficiency manifested with elevated levels of inactive, acetylated PGC-1α and mitochondrial dysfunction, as evidenced by reduced basal oxygen consumption, fewer functional and total mitochondria per hepatocyte, and decreased levels of mitochondrially encoded cytochrome C oxidase I (MTCO1) in complex IV and mitochondrially encoded ATP Synthase membrane subunit 6 (MTATP6) in complex V. There was also an increase in ATP citrate lyase (ACLY) activity and acetyl-CoA levels, which would further promote the acetylation-based inactivation of PGC-1α. Crucially, hepatic overexpression of Sirtuin-1 (SIRT1), an NAD+-dependent deacetylase that can deacetylate and activate PGC-1α, lowered and normalized acetyl-PGC-1α levels, restored the expression of ETC components, and increased mitochondrial complex IV activity. These results suggest that down-regulation of SIRT1-PGC-1α signaling is a key factor in the mitochondrial dysfunction observed in GSD Ia [88].

### 3.3. Autophagy and Mitophagy

Macroautophagy, commonly known as autophagy, is an evolutionarily conserved process that supplies building blocks, such as amino acids and glucose, originating from the degradation of macromolecules. These building blocks can be used as energy fuel during periods of starvation. In the absence of nutrients, cells utilize their internal resources by enclosing portions of the cytoplasm in autophagic vesicles (autophagosomes) [89,90]. These vesicles then transport their contents to lysosomes for degradation and recycling. There are also organelle-specific autophagic pathways, serving as a quality control system for recycling compromised organelles. The subclass of autophagy which recycles damaged mitochondria is called mitophagy. This quality control pathway can restore mitochondrial function once their repair is no longer possible. Under these circumstances, the entire organelle is removed via the mitophagic pathway, maintaining cellular homeostasis. Mitophagy plays a crucial role in post-mitotic cells such as neurons, which cannot increase glycolytic ATP production as compensation for OxPhos deficiency. Eliminating damaged mitochondria is essential to prevent the buildup of ROS, which inevitably results from dysfunctional OxPhos. Accumulation of ROS can further impair mitochondrial function, thereby compromising the cell’s energy homeostasis [91,92]. Mitophagy begins with the accumulation of PINK1, a serine-threonine protein kinase, on the outer mitochondrial membrane (OMM) when membrane potential (ΔΨm) is lost (Figure 5). PINK1 then signals mitochondrial dysfunction to PARK2, a cytosolic E3 ubiquitin ligase, which facilitates the autophagosomal engulfment of damaged mitochondria, leading to their degradation in lysosomes. In GSDs, impaired autophagy can lead to the accumulation of dysfunctional mitochondria, exacerbating oxidative stress and leading to cellular damage [93,94]. Downregulation of the pro-autophagic AMPK signaling and up-regulation of anti-autophagic mTOR signaling were found to be associated with impaired autophagy. Several signaling pathways, including the AMPK-mTOR pathway, regulate mitophagy and play a cardioprotective role by clearing out abnormal mitochondria, thereby preventing oxidative stress and reducing apoptosis in cardiomyocytes [3,95,96]. Autophagy was induced using either ULK1 overexpression or by rapamycin treatment to inhibit mTOR, leading to decreased hepatocyte glycogen and lipid stores in neonatal G6pc knock-out mice, as well as G6PC-deficient dog model of GSD la [97]. In contrast, subsequent research by Cho et al., in adult liver-targeted G6pc knockout mice, indicated that the down-regulation of SIRT1 and its target gene, FOXO1, played a crucial role in the suppression of autophagy. Restoration of SIRT1 signaling re-established autophagy but failed to correct other metabolic anomalies, suggesting that pathways beyond SIRT1 regulation of autophagy are likely disrupted in GSD Ia [98]. Additionally, this model exhibited minimal dysfunction in mTOR signaling, possibly preventing rapamycin treatment from reinstating autophagy [98]. Discrepancies in model type and the age of mice used might explain the variations in mTOR signaling observed across these studies. Further investigation in the adult liver-specific G6pc knockout mice suggested that an increased flux of metabolites through the hexose-monophosphate shunt could influence SIRT1 signaling and reduce hepatic autophagy [98]. Impaired autophagy may also result in diminished mitochondrial function, as mitochondria damaged by oxidative stress are unable to undergo removal via mitophagy [99]. Lafora disease is a glycogen storage disorder caused by mutations in either laforin or malin, in which cells accumulate polyglucosan bodies (branched polymers of glucose) called Lafora bodies in tissues such as brain, heart, liver, muscle, and skin [100]. Cell culture studies have demonstrated impaired mitophagy in fibroblasts lacking laforin suggesting mitochondria could still be targeted for degradation but could not progress further to mitophagy [101]. Mitochondrial dysfunction due to excessive ROS activity and oxidative stress and flawed autophagy are common features of many GSDs. A study of mouse and human models of PD identified several mitochondrial defects in cardiac and skeletal muscle myopathy, including profound dysregulation of Ca^2+^ homeostasis, mitochondrial Ca^2+^ overload, increased ROS production, decreased mitochondrial membrane potential, elevated caspase-independent apoptosis, as well as reduced oxygen consumption and ATP production in mitochondria. This study concluded that disturbances in Ca^2+^ homeostasis and mitochondrial abnormalities are early pathogenic changes in PD [70]. 

### 3.4. Apoptosis and Mitochondrial Dysfunction

Apoptosis is a highly regulated form of cell death closely associated with mitochondrial function, which plays a critical role in mitochondrial-associated neurodegenerative diseases [102]. Mitochondria are not limited to energy production through ATP synthesis but also to integrating essential cellular signaling pathways. In apoptosis, damaged mitochondria can no longer maintain the proper electrical potential across mitochondrial membranes, thus releasing cytochrome c into the cytosol, where it binds to proteins that form the apoptosome, activating caspases that execute the cell program death [103,104]. The apoptotic pathway relies on caspase activation within an apoptosome, consisting of an oligomeric caspase activator (CED-4, Dapaf-1, Apaf-1) and CARD-carrying initiator caspases (CED-3, DRONC, Caspase-9), with cytochrome c as a crucial co-factor. Bcl-2 family proteins, either proapoptotic (e.g., EGL-1, Bax) or antiapoptotic (e.g., CED-9, Bcl-2), regulate the release of cytochrome c and other apoptogenic factors from the mitochondria, thereby controlling apoptosome activation and apoptosis initiation (Figure 6) [105]. Mitochondrial Ca^2+^ overload can lead to necrosis or apoptosis [70]. An increased Ca^2+^ level causes mitochondrial abnormalities, which in turn contribute to muscle damage and wasting through mitochondria-mediated apoptosis. In G6PC KD cells, we found a striking decrease in membrane potential. It was associated with the translocation of cytochrome c into the cytoplasm and cleavage and activation of caspase 9, a specific marker of the mitochondrial apoptosis pathway [12] that can cause hepatocellular apoptosis one of the important clinical features of GSD la [106]. Autophagy and apoptosis are two distinct yet interconnected pathways that significantly influence cell fate. Autophagy is essential for cell survival during periods of starvation and acts as a cellular cleanup mechanism under var ious stress conditions. In contrast, apoptosis is a form of programmed cell death crucial for maintaining cellular homeostasis and eliminating damaged or unnecessary cells. Decreased mitochondrial membrane potential, reactive oxygen species (ROS) production, calcium overload, and apoptosis indeed precede the development of autophagic buildup and lipofuscin accumulation in PD [70].

### 3.5. Mitochondrial Myopathy

All GSDs (except for GSDs type 1 and 6, which are primarily hepatopathies) are associated with myopathies and are well-linked to mitochondrial dysfunction. These myopathic GSDs are characterized by muscle weakness and fatigue due to impaired energy production in muscle cells. The accumulation of glycogen and defective mitochondrial function in muscle tissues contribute to these symptoms. Muscle weakness and exercise intolerance are common clinical features of several GSDs, often associated with mitochondrial dysfunction [107]. In GSD V, muscle phosphorylase deficiency leads to a lack of glucose availability during exercise, impairing ATP production and causing muscle fatigue [14]. Similarly, in PD, the accumulation of glycogen in muscle cells presumably compromises mitochondrial function, leading to muscle weakness and respiratory difficulties [108]. Skeletal muscle is highly energy-dependent and, therefore, particularly susceptible to disorders affecting energy metabolism. ATP is the immediate and essential source of energy for both contraction and relaxation of muscle fibers. To regenerate ATP, skeletal muscle uses various substrates, including high-energy phosphate compounds, glucose, glycogen, which turns over to glucose, and free fatty acids (FFAs). This ATP regeneration occurs through multiple metabolic pathways, such as the creatine kinase (CK) reaction, anaerobic glycolysis, the β-oxidation spiral, and OxPhos. At rest, fatty acids (FAs) are the primary source of fuel for skeletal muscle [109]. However, high-intensity exercise relies more on anaerobic glycolysis, the metabolism of glucose to produce ATP. OxXPhos is the main mechanism of producing ATP during submaximal exercise. OxPhos uses NADH, derived by the TCA cycle from metabolized glycogen (through glucose), glucose, and FAs. These reducing equivalent (NADH and FADH_2_) donate electrons in the mitochondrial electron transport chain to produce ATP. Defects in ATP production might cause muscular disorders associated with impairment of specific metabolic pathways, depending on which substrates are used as a fuel source. Mitochondrial respiratory chain complexes are essential components of the mitochondrial respiratory chain and play a major role in generating ATP. In GSD V, the enzyme myophosphorylase catalyzes the rate-limiting step in muscle glycogen metabolism by releasing glucose-1-phosphate from terminal alpha-1,4-glycosidic bonds. Due to a deficiency of this enzyme, muscle fibers cannot obtain energy from intracellular glycogen stores, leading to impaired glycolytic flux. Impaired muscle aerobic metabolism is a hallmark of GSD V characterized by very low levels of peak oxygen uptake (VO_2_ peak) [110,111]. GSD V (McArdle) patients were evaluated by exercise testing, and their VO_2_ peak levels were found to be much lower than in healthy controls, consistent with impaired oxidative metabolism [112,113,114]. Muscle biopsy analyses from two patients revealed loss of mitochondrial and cytoskeletal integrity, particularly in type II fibers, suggesting that disruption of the mitochondrial network may contribute to the reduced muscle oxidative capacity in this disease [14]. In a study on GSD type 5 (McArdle disease), an impairment in mitochondrial content and biogenesis, a decrease in OxPhos complex proteins and activities, together with high muscle NADH but reduced muscle levels of glucose-6-phosphate and of the glycolytic products pyruvate and lactate were found (Figure 7). The authors suggest that the deficiency in oxidative metabolism in McArdle disease might be caused by reduced levels of OxPhos substrates as a result of reduced glycogen degradation and a disruption of the mitochondrial network [14]. Exercise intolerance in McArdle disease arises from two main mechanisms: the block of anaerobic glycolysis deprives muscles of energy for isometric exercise, and the block of aerobic glycogen utilization reduces pyruvate and acetyl-CoA, impairing dynamic exercise. Impaired OxPhos resulted in decreased oxygen consumtion and maximal oxygen uptake, as reported in McArdle patients [115]. Muscle glycogen synthase (GYS1) deficiency, implicated in both skeletal and cardiac muscles is a characteristic feature of GSD 0b. GSD 0b patients present with a significant predominance of type I (oxidative) muscle fibers and mitochondrial proliferation together with selective atrophy of type II (glycolytic) fibers. These clinical observations suggest inability to mobilize glycogen into glycolytic fuel with compensatory up-modulation of mitochondrial OxPhos metabolism [17]. Glycogenin-1, encoded by the GYG1 gene, is an enzyme initiating glycogen biosynthesis., Mutations in glycogenin-1 were described in GSD XV. Muscle biopsies taken from GSD XV patients showed reduced glycogen, mitochondrial proliferation, and type I fiber predominance [116]. A case report in 7 patients with glycogenin-1 deficiency showed progressive proximal weakness, a myopathic EMG, and polyglucosan bodies in the muscle biopsy [42]. Hypertrophic cardiomyopathy is a key feature of PD manifested by the marked thickening of the ventricular walls and associated hyperdynamic systolic function with outflow tract obstruction. Disturbed metabolic signaling and mitochondrial dysfunction are common pathogenic mechanisms of hypertrophic cardiomyopathy [67]. Therefore, it is likely that mitochondria significantly contribute to the development of cardiac hypertrophy in PD patients. Mitochondrial dysfunction was found to be associated with aberrant energy metabolism in skeletal muscle biopsy of PD patients and PD animal models [70]. Huang et al. reported the presence of swollen cristae and mitochondrial dysfunction, including decreased glycolysis and OxPhos in iPSC-derived from the PD patients’ fibroblasts [69] In addition, deformed mitochondria were observed in muscle biopsy and exhibited a significant decrease in the number of mitochondria, in consumption of oxygen, and in ATP production, as well as a collapse in mitochondrial membrane potential with increased levels of ROS [70]. These findings are in support of the significant role mitochondrial dysfunction and impairment migt play in the development of myopathy in GSDs, however, further research is needed to explore this mechanism fully.

## 4. Diagnosis and Therapeutic Strategies in GSDs

### 4.1. Diagnosis

Based on experts’ opinions, clinical practice guidelines have been developed for diagnostic and therapeutic purposes, including most GSD types [11,24,32,66,117]. Newborn screening (NBS) is a good option in the diagnosis of GSDs with certain limitations requiring muscle or liver tissue for functional assays in many GSDs. It often requires metabolite detection, enzyme activity measurements, or genetic tests using dried blood specimens [118]. NBS is very common in PD using enzymatic assay [119]. The diagnosis of mitochondrial dysfunction in glycogen storage disorders requires an integrative approach combining clinical, biochemical, imaging, histological, and genetic analyses. This comprehensive evaluation helps to accurately diagnose the underlying disorder and guide appropriate management and treatment strategies. Detailed patient history, including a family history of metabolic disorders and physical examination to identify symptoms such as muscle weakness, exercise intolerance, hepatomegaly, and cardiomyopathy, will be useful. Laboratory tests include measuring blood glucose levels, lactate, pyruvate, amino acids, and acylcarnitines. Elevated lactate and pyruvate levels may indicate mitochondrial dysfunction. Liver function tests and creatine kinase (CK) levels can help identify liver involvement and muscle damage [66,120,121]. Specific enzyme activity assays in muscle or liver biopsy samples to identify deficiencies related to GSDs (e.g., glycogen synthase, debranching enzyme, branching enzyme). Imaging tests, including magnetic resonance imaging (MRI) and ultrasonography (USG) for liver and abnormalities in muscle texture and composition, echocardiography is important for individuals at risk of cardiomyopathy and arrhythmias such as those with GSD l, PD, GSD III or GSD IV. Histological examination is used to identify abnormal glycogen accumulation and structural changes, and Electron Microscopy can provide detailed images of mitochondrial structure to find out morphological abnormalities. The genomic analysis includes next-generation sequencing (NGS), targeted gene panels, and whole exome sequencing (WES) to identify mutations and sequencing of nuclear and mitochondrial genes. This can provide a confirmatory and clear picture of the association of mitochondrial dysfunction with GSDs. Mitochondrial respiratory chain and OXPHOS studies in which measurement of the activities of respiratory chain complexes (I-IV) and oxygen consumption rates (OCR) in cells or isolated mitochondria to assess overall mitochondrial function have been recommended in studies [122,123,124]. 

### 4.2. Current Treatment Options and Therapies

Enzyme replacement therapy (ERT) has been successfully developed for some GSDs, mainly PD. ERT aims to replace the deficient enzyme, thus reducing glycogen accumulation and improving mitochondrial function. Clinical studies have shown that ERT can improve muscle strength, respiratory function, and cardiac abnormalities in PD, highlighting its potential to mitigate mitochondrial dysfunction [39,125]. Gene therapy offers a promising approach to correcting the underlying genetic defects in GSDs. By delivering functional copies of the deficient gene, gene therapy can restore enzyme activity, reduce glycogen accumulation, and improve mitochondrial function. Advanced gene editing techniques, such as CRISPR-Cas9, offer the potential to precisely correct genetic mutations in GSDs. By targeting the specific mutations responsible for enzyme deficiencies, gene editing could restore normal enzyme activity and improve mitochondrial function. Preclinical studies have shown encouraging results in animal models of GSDs, and clinical trials are underway to evaluate the safety and efficacy of gene therapy in humans [126,127,128]. Given the role of oxidative stress in mitochondrial dysfunction, antioxidant therapy has been explored as a potential treatment for GSDs. Antioxidants, such as coenzyme Q10, vitamin E, and N-acetylcysteine, can reduce ROS levels and protect mitochondrial function [129,130]. Clinical studies have shown that antioxidant therapy can improve muscle function and reduce oxidative stress in patients with GSDs. A study reported that nutritional co-therapy with 1,3-butanediol and multi-ingredient antioxidants may provide an alternative to ketogenic diets for inducing ketosis and enhancing autophagic flux in PD [131]. Metabolic modulation involves altering metabolic pathways to improve energy production and reduce glycogen accumulation. For instance, ketogenic diets, which are high in fat and low in carbohydrates, can promote fatty acid oxidation and ketone body production, providing an alternative energy source for mitochondria. Clinical studies have shown that ketogenic diets can improve exercise tolerance and reduce muscle symptoms in patients with GSD V. Mitochondrial-targeted therapies aim to improve mitochondrial function directly. These therapies include compounds that enhance mitochondrial biogenesis, such as peroxisome proliferator-activated receptor gamma coactivator 1-alpha (PGC-1α) agonists, and agents that improve mitochondrial dynamics, such as mitochondrial division inhibitor 1 (Mdivi-1). A preclinical study by our group explores the efficacy and mechanism of action of the polyglucosan-reducing compound 144DG11 (GHF 201) in GBE knockin (Gbe^ys/ys^) mouse model and patient skin fibroblast of adult polyglucosan body disease (APBD). Moreover, 144DG11 therapy can improve mitochondrial function and reduce disease symptoms in animal models of GSDs. Lysosomal membrane protein LAMP1 is the molecular target of 144DG11, which enhanced autolysosomal glycogen degradation by increasing autophagic flux and lysosomal acidification. Moreover, 144DG11 increases carbohydrate burn at the expense of fat burn, suggesting metabolic mobilization of pathogenic polyglucosan, a key feature of APBD, and increases glycolytic, mitochondrial, and total ATP production [23]. Empagliflozin, an inhibitor of the renal sodium-glucose cotransporter type 2 (SGLT2), improves redox state and oxidative stress, inhibiting reactive oxygen species (ROS) production, reducing the activity of pro-oxidant agents, and improving mitochondrial function [132]. Studies reported that empagliflozin improved symptoms like neutropenia, hypoglycemia, and inflammatory bowel syndrome in GSD Ib patients [133,134,135] (Table 2).

## 5. Conclusions

Mitochondrial dysfunction in glycogen storage disorders (GSDs) represents a critical aspect of these metabolic diseases, underscoring the complex interplay between cellular energy management and glycogen metabolism. GSDs, characterized by deficiencies in enzymes involved in glycogen synthesis or degradation, lead to the accumulation or improper utilization of glycogen in tissues such as the liver and muscle. This metabolic dysregulation often results in impaired energy production within mitochondria. Studies have shown that mitochondrial dysfunction in GSDs manifests through various mechanisms including altered mitochondrial biogenesis, disturbed ROS activity, increased oxidative stress, and impaired OXPHOS. These anomalies resulted in impaired structure and function of the mitochondria and contributed to clinical symptoms such as muscle weakness, exercise intolerance, and hepatic dysfunction, which are very common in GSDs. Furthermore, the intricate relationship between mitochondrial function and glycogen metabolism suggests that targeting mitochondrial pathways could offer therapeutic potential for managing GSDs. Advancements in molecular biology and genetics have provided deeper insights into the mitochondrial disturbances in GSDs, highlighting the need for comprehensive diagnostic and therapeutic strategies that address both glycogen metabolism and mitochondrial health. Interventions aiming to restore mitochondrial function, such as antioxidant therapy, gene therapy, and enzyme replacement therapy, hold promise but require further research and clinical validation. In conclusion, mitochondrial dysfunction plays a pivotal role in the pathophysiology of glycogen storage disorders, significantly influencing disease outcomes and patient quality of life. A multidisciplinary approach that integrates metabolic, genetic, and mitochondrial-targeted therapies is essential for developing effective treatments for GSDs, ultimately aiming to improve clinical outcomes and enhance the well-being of affected individuals.

## Figures and Tables

**Figure 1 biomolecules-14-01096-f001:**
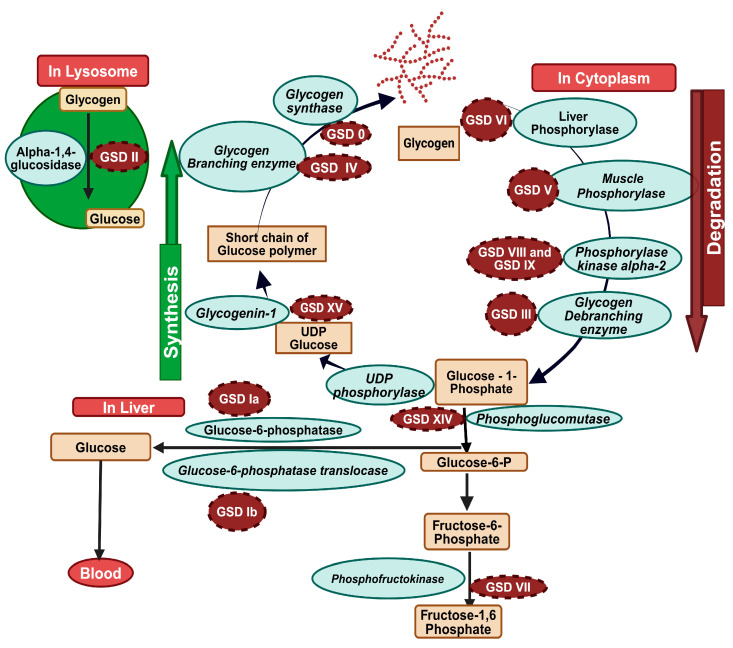
Glycogen metabolism pathway’s enzymes defects and GSDs; (defective enzymes are in cyan color and resulting GSDs are in dark brown circle, metabolism products are shown in light brown color (created with BioRender.com, accessed on 8 August 2024).

**Figure 2 biomolecules-14-01096-f002:**
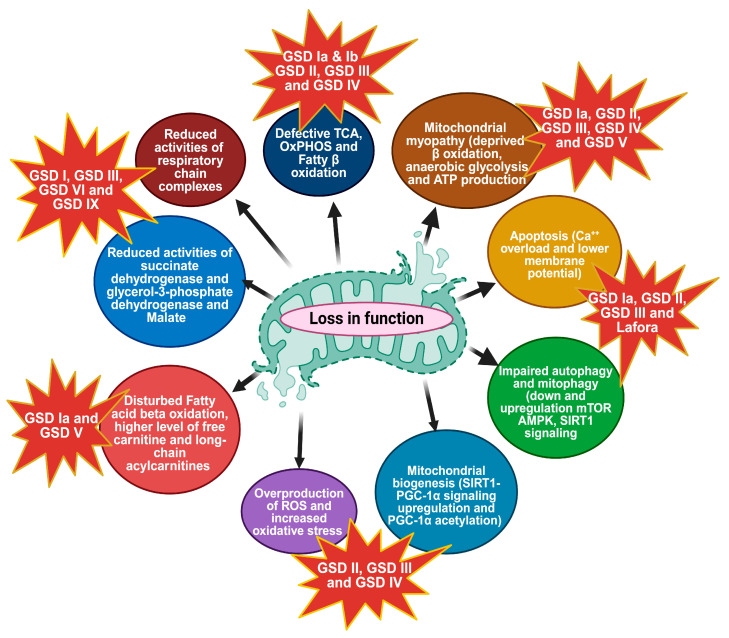
Major mitochondrial dysfunctions in glycogen storage disorders (GSDs), (created with BioRender.com, accessed on 17 August 2024).

**Figure 3 biomolecules-14-01096-f003:**
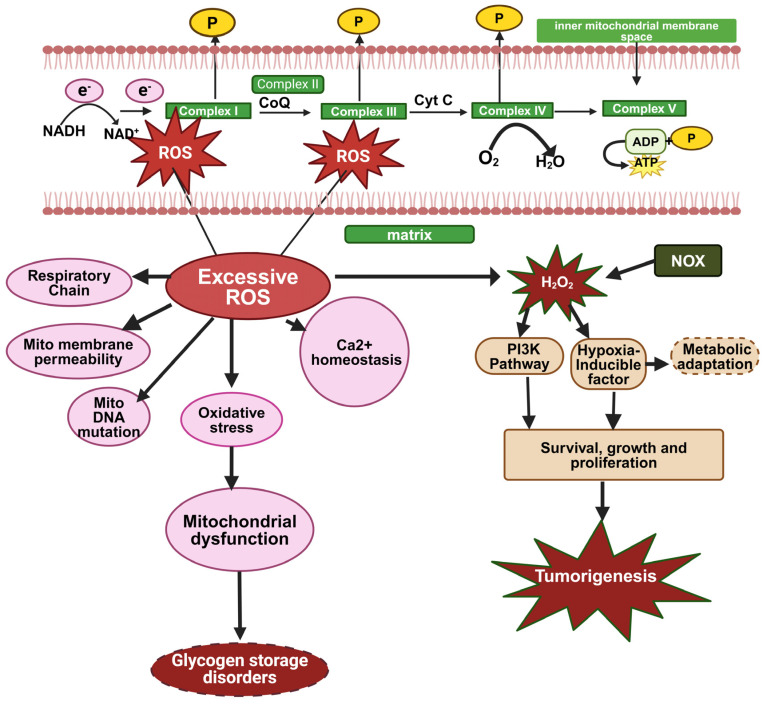
Mitochondrial dysfunction and tumorigenesis by excessive ROS formation, (created with BioRender.com, accessed on 17 August 2024).

**Figure 4 biomolecules-14-01096-f004:**
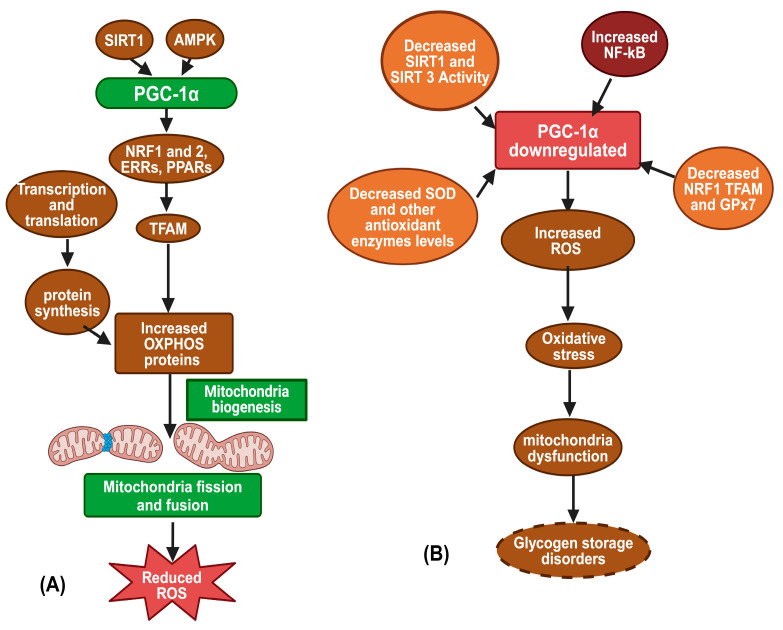
(**A**) Mitochondria biogenesis (**B**) PGC-1α downregulation impaired Mitochondrial ROS defense (created with BioRender.com, accessed on 27 August 2024).

**Figure 5 biomolecules-14-01096-f005:**
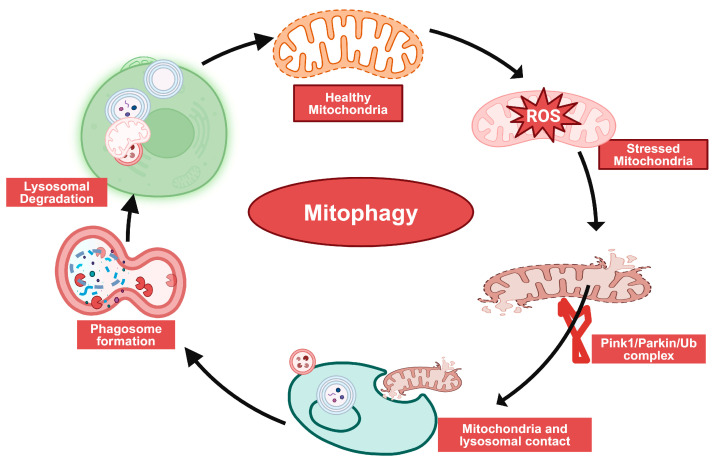
Mitophagy starts with stressed and dysfunctional mitochondria; impaired mitochondria loss membrane potential, PINK1/parkin RBR E3 ubiquitin protein ligase complex formation, phagophore formation and lysosomal degradation (created with BioRender.com, accessed on 17 August 2024).

**Figure 6 biomolecules-14-01096-f006:**
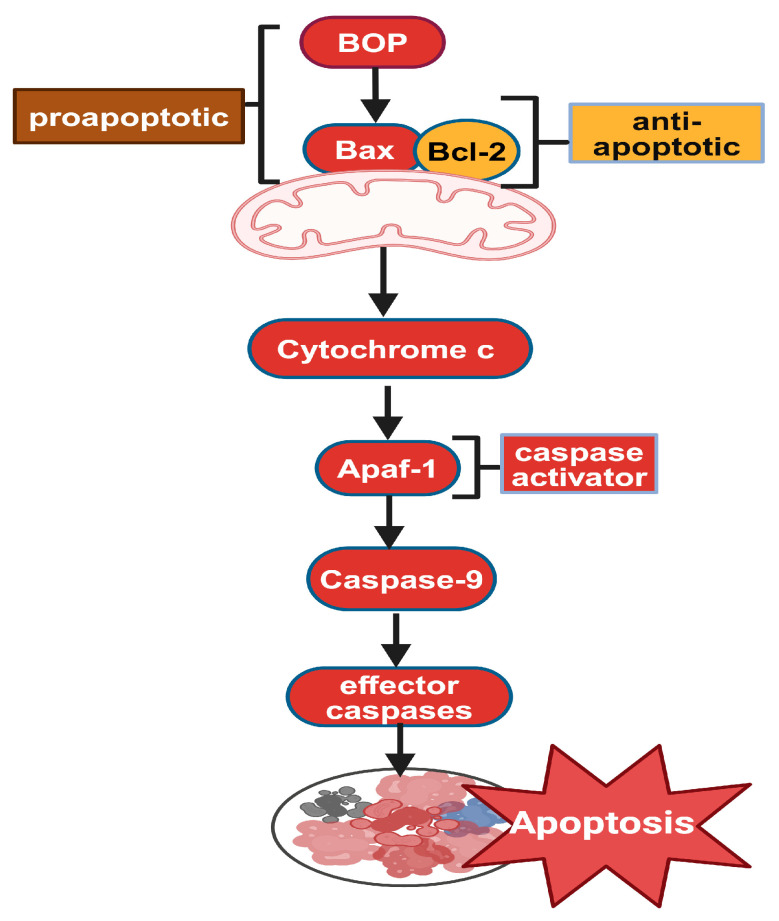
Mitochondrial apoptotic pathway (created with BioRender.com, accessed on 22 August 2024).

**Figure 7 biomolecules-14-01096-f007:**
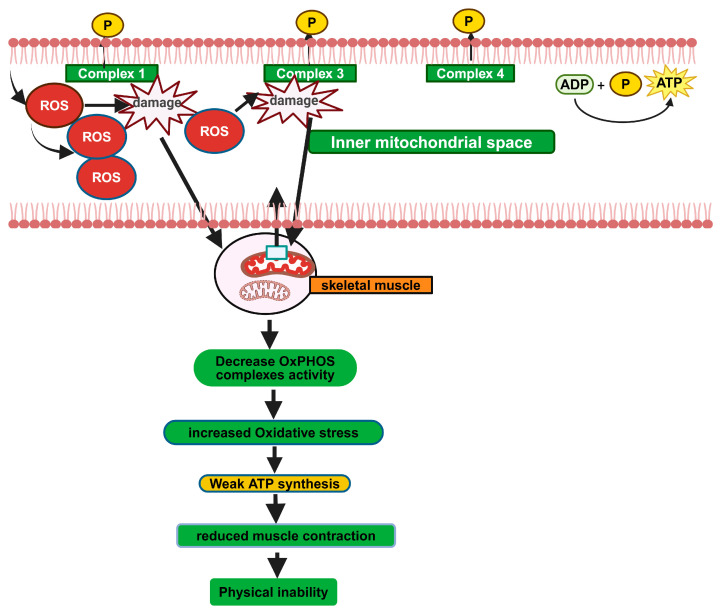
OxPHOS complex proteins imbalance and myopathy (created with BioRender.com, accessed on 21 August 2024).

**Table 1 biomolecules-14-01096-t001:** Enzyme defects with clinical features and epidemiology in GSDs.

Disorders (Common Name)	Enzyme	Gene/Chromosome/-Inheritance	Clinical Manifestation
GSD 0a	Glycogen synthase in Liver	GYS2/12p12.1/AR	Fasting ketotic hypoglycemia, reactive hyperglycemia and lactate elevation [15,16]
GSD 0b	Glycogen synthase in muscle	GYS1/19q13.33/AR	Hypertrophic cardiomyopathy exercise intolerance, and adult-onset myopathy without cardiomyopathy [17,18]
GSD Ia (von Gierke disease)	Glucose-6-phosphatase	G6PC/17q21.31/AR	Hypoglycemia, lactic acidosis, Hypertriglyceridemia, hepatomegaly, renal dysfunction [19]
GSD Ib	Glucose-6-phosphate translocase	SLC37A4/11q23.3/AR	Neutropenia, neutrophil dysfunction and inflammatory bowel disease [20]
GSD II (PD)	alpha-1,4-glucosidase	GAA/17q25.3/AR	Hypertrophic cardiomyopathy, hypotonia, and motor delay [21]
GSD III (Cori or Forbes disease)	Glycogen debranching (amylo-1,6 glucosidase)	AGL/1p21.2/AR	Hypoglycemia, elevated ketosis in hyperlipidemia, hepatomegaly, elevated liver enzyme myopathy, variable muscle and cardiac phenotype [11]
GSD IV (Andersen or Adult polyglucosan body disease)	Glycogen branching enzyme- amylo (1,4-1,6) transglucosidase	GBE1/3p12.2/AR	Hepatosplenomegaly, liver dysfunction, progressive cirrhosis, cardiomyopathy, hypotonia, gait difficulty, progressive neurogenic bladder, autonomic dysfunction, sensory loss, and variable cognitive difficulty [22,23,24]
GSD 5 (McArdle disease)	Myophosphorylase	PYGM/11q13.1/AR	Exercise induces fatigue, cramps, tachypnoea and tachycardia, rhabdo-myolysis, myoglobinuria [25]
GSD 6 (Hers disease)	Liver glycogen phosphorylase	PYGL/14q22.1/AR	Hepatomegaly, hypoglycemia, with ketosis, elevated liver transaminases, hyperlipidemia, osteoporosis, and liver fibrosis [26,27]
GSD 7 (Tarui disease)	Muscle phosphofructokinase	PFKM/12q13.11/AR	Hemolytic anemia, muscle weakness, exercise-induced muscle cramping, exertional Myopathy, and gout/hyperuricemia [28,29]
GSD 9A1 (formerly GSD 8)	Alpha-2 subunit of liver phosphorylase kinase	PHKA2/Xp22.13 XL	Hepatomegaly, growth retardation, motor developmental delay. Hypercholesterolmia, hypertriglyceridemia, elevated liver enzymes; fasting hyperketosis [30]
GSD 9B (GSD IXb)	Beta subunit of liver and muscle phosphorylase	PHKB/16q12.1/AR	Short stature, hepatomegaly, diarrhea, muscle weakness, hypotonia [3,31]
GSD 9C (GSD IXc)	Hepatic and testis isoform gamma subunit of phosphorylase kinase	PHKG2/16p11.2/AR	Growth retardation, hepatomegaly, hypotonia; cognitive delay [32,33]
GSD 9D (GSD IXd)	Alpha subunit of muscle phosphorylase kinase	PHKA1/Xq13.1/XL	Muscle weakness, exercise-induced muscle pain and stiffness, muscle atrophy, elevated CK [3,30,32]
GSD 10 (GSD X)	Muscle phosphoglycerate mutase	PGAM2/7p13/AR	Exercise intolerance with cramp or pain, rhabdo-myolysis, myoglobinuria, hyperuricemia, coronary arteriosclerosis [2,34]
GSD 11 (GSD XI)	Lactate dehydrogenase A	LDHA/11p15.1/AR	Exercise-induced muscle cramps and pain, uterine muscle stiffness in pregnancy, psoriatic skin lesions, Elevated serum CK during myoglobinuria, with low serum lactate dehydrogenase [35,36,37]
GSD 12 (GSD XII)	Fructose-1,6- bisphosphate aldolase	ALDOA/16p11.2/AR	Short stature dysmorphic face, myopathy; mental retardation; delayed puberty; hemolytic anemia, hepatosplenomegaly; rhabdomyolysis with febrile illness [38,39]
GSD 13 (GSD XIII)	Beta-enolase	ENO3/17p13.2/AR	Exercise intolerance, myalgia, rhabdomyo-lysis with fatty infiltration [40,41]
GSD 15 (GSD XV)	Glycogenin-1	GYG1/3q24/AR	Weakness, arrhythmias, skeletal myopathy, cardiomyopathy [42,43]
GSD 14 (Previously GSDXIV)	Phosphoglucomutase-1	PGM1/1p31.3/AR	Hematological anomalies, hypoglycemia, growth retardation, and dilated cardio-myopathy [44,45]
Fanconi–Bickel syndrome (GSD XI)	SLC2A2	GLUT2/3q26.2/AR	Postprandial elevations of glucose and galactose, fasting hypoglycemia, hepatomegaly, proximal tubular nephropathy, glucosuria, short stature [46,47]
PGK deficiency	PGK1	PGK/Xq21.1/XL	Nonspherocytic hemolytic anemia, myopathy with rhabdomyolysis and neurological features, myopathy, rhabdomyolysis [48,49]

AR = autosomal recessive; XL = X-linked.

**Table 2 biomolecules-14-01096-t002:** Current diagnosis and treatment/management option for common GSDs.

Type of GSDs	General Lab Test and Imaging	Diagnostic Test	Genetic Test	Treatment and Management
GSD 0a	NA	NA	Biallelic PLP GYS2 variants	Protein-rich meals and bedtime consumption of uncooked cornstarch
GSD 0b	NA	NA	Biallelic PLP GYS1 variants	
GSD I	Kidney and liver function tests, lipids and uric acid levels, a complete blood count, and iron studies. Liver and kidney imaging, echocardiography	GSD Ia, G6Pase enzyme activity in liver tissue	Biallelic PLP G6PC1 variants (GSD Ia), biallelic PLP SLC37A4 (GSD Ib)	Angiotensin receptor blockers, low-purine diet, and Allopurinol for gout, G-CSF, and empagliflozin Liver transplantation
GSD II	CK, AST, ALT, urine Glc4, BNP, and, when receiving ERT, IgG against recombinant protein levels. ECG, Echo, Chest X-Ray	GAA activity testing in blood, fibroblast, or muscle	Biallelic PLP variants in GAA	ERT, Respiratory muscle training, maximize clearance of airway secretions
GSD III	Liver function tests, coagulation, Lipid, CK levels, echocardiography	GDE activity assay for liver and muscle, can be measured in leukocytes, erythrocytes, or cultured fibroblasts	Biallelic PLP AGL variants	Dietary management, hypoglycemia monitoring and muscle health support
GSD IV	liver function tests, PT/INR, albumin, renal function panel, complete blood count, ammonia levels, AFP levels, USG, Bone density scan, physical therapy/occupational therapy	GBE activity assay in on fibroblast, liver or muscle and PB accumulation by PAS staining	Biallelic PLP GBE1 variants	Hepatitis A and B vaccinations. Prophylactic antibiotics for small bowel infections and urinary tract infections. Physical therapy and occupational therapy. Routinely screen for dysphagia (difficulty swallowing). Liver transplant if required
GSD 5	Serum CK and uric acid levels, Hb A1c, lipid profile. weakness and muscle evaluation. Assess ADLs and QoL	Myophosphorylase activity	Biallelic PLP PYGM	Carbohydrate rich diet, Sucrose to improve exercise tolerance. Exercise is helpful for chronic pain
GSD 6	AST, ALT, serum albumin and γ-glutamyl transferase levels, coagulation, blood glucose, and serum β-hydroxybutyrate monitoring	Phosphorylase activity in liver specimens	Biallelic PLP variants in PYGL	Dietary and symptomatic treatment
GSD 7		Phosphofructokinase activity in muscle	Biallelic PLP PFKM variants	Low-carbohydrate ketogenic diet. Exercise to avoid chronic pain
GSD 9	AST, ALT, serum albumin and γ-glutamyl transferase levels, coagulation DEXA, and heart imaging. Blood glucose and serum β-hydroxybutyrate monitoring	Phosphorylase b kinase activity can be measured in liver, blood, and muscle (based on subtype). Enzyme activity in blood can be normal	Hemizygous PLP variant in PHKA2 (GSD IX A) biallelic PLP variants in PHKB (GSD IX B)	Dietary and symptomatic treatment

G6Pase (glucose-6-phosphatase); GAA (acid α-glucosidase); GDE (glycogen debranching enzyme); PAS (periodic acid–Schiff); PLP (pathogenetic or likely pathogenetic); ADL (activities of daily living); AFP (α-fetoprotein); ALT (alanine aminotransferase); AST (aspartate aminotransferase); CK (creatine kinase); DEXA (dual X-ray absorptiometry); ERT (enzyme replacement therapy); G-CSF (granulocyte colony-stimulating factor); Glc4 (glucose tetrasaccharide); PT/INR (prothrombin time/international normalized ratio); PFKM (Phosphofructokinase, Muscle); PHKA2 (phosphorylase kinase regulatory subunit alpha 2); PHKB (Phosphorylase b kinase enzyme); QoL (quality of life); NA (not available).

## Data Availability

No new data were created or analyzed in this study. Data sharing does not apply to this article.

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
