# Peer review of "Mitochondrial Dysfunction in Glycogen Storage Disorders (GSDs)"

_biomolecules, 2024, doi:10.3390/biom14091096_

Round 1
Reviewer 1 Report
Comments and Suggestions for Authors
Mitochondrial dysfunction can be a feature of glycogen storage disorders (GSDs), which are a group of inherited metabolic disorders that affect enzymes involved in glycogen metabolism. Deficiencies in enzymes that break down and synthesize glycogen can impair mitochondrial function. In their submitted manuscript, Kumudesh and Or Kakhlon summarized the intricate relationship between mitochondrial dysfunction and various types of GSDs, which is valuable. I have the following questions/comments.
1. Please make a figure to summarize the ‘4. Mitochondrial dysfunction targeted diagnosis and therapeutic strategies in GSDs’.
2. If possible, please make a figure or a table to summarize the ‘3.1. Overproduction of ROS and oxidative stress’
3. If possible, please make a figure or a table to summarize the ‘3.2 Mitochondrial biogenesis’.
4. If possible, please make a figure or a table to summarize the ‘3.3 Autophagy and mitophagy.
5. If possible, please make a figure or a table to summarize the ‘3.4. Apoptosis and mitochondrial dysfunction.
6. If possible, please make a figure or a table to summarize the ‘3.5. Mitochondrial myopathy’.
7. Some modifications/corrections in the manuscript may be needed.
Comments on the Quality of English Language
Good.
Author Response
Reviewer 1 comments
Dear Reviewer,
We sincerely appreciate your thorough review of our manuscript and the valuable comments you provided. In response, we have addressed all your queries with point-by-point clarifications (in bold letters). All the modifications have been highlighted in yellow within the main text of the manuscript.
Comments and Suggestions for Authors
Mitochondrial dysfunction can be a feature of glycogen storage disorders (GSDs), which are a group of inherited metabolic disorders that affect enzymes involved in glycogen metabolism. Deficiencies in enzymes that break down and synthesize glycogen can impair mitochondrial function. In their submitted manuscript, Kumudesh and Or Kakhlon summarized the intricate relationship between mitochondrial dysfunction and various types of GSDs, which is valuable. I have the following questions/comments.
- Please make a figure to summarize the ‘4. Mitochondrial dysfunction targeted diagnosis and therapeutic strategies in GSDs’.
Currently, no specific diagnostic or treatment options for glycogen storage diseases (GSDs) directly target mitochondrial dysfunction. However, we have summarized the current potential diagnostic and treatment approaches for the most significant GSDs in Table 2. We chose a table format instead of a figure to ensure clarity and ease of understanding. We changed the heading 4. “Mitochondrial dysfunction targeted diagnosis and treatment” to “Current diagnosis and therapeutic strategies in GSDs”.
- If possible, please make a figure or a table to summarize the ‘3.1. Overproduction of ROS and oxidative stress’
We have prepared Figure 3 for the above condition as suggested
- If possible, please make a figure or a table to summarize the ‘3.2 Mitochondrial biogenesis’.
I apologize, we can’t prepare due to the excess number of figures
- If possible, please make a figure or a table to summarize the ‘3.3 Autophagy and mitophagy.
We have prepared Figure 4 for the above condition as suggested
- If possible, please make a figure or a table to summarize the ‘3.4. Apoptosis and mitochondrial dysfunction.
I apologize, we can’t prepare due to the excess number of figure
- If possible, please make a figure or a table to summarize the ‘3.5. Mitochondrial myopathy’.
I apologize, we can’t prepare due to the excess number of figures
Some modifications/corrections in the manuscript may be needed.
We have gone through the manuscript and made all required modifications/corrections.
Comments on the Quality of English Language- Good.
Thank you
Best Regards
Kumudesh

Reviewer 2 Report
Comments and Suggestions for Authors
General comments:
The paragraphs are extremely long and therefore difficult to follow. Shorter paragraphs with a logical progression are advised. It is well known that glycogen storage disorders are associated with mitochondrial dysfunction. This review should have explore how reduced glycogen synthesis or abnormal degradation impacts on the energy provision pathways in the cell and suggest how this can lead to functional abnormalities in the mitochondrion.
Specific comments:
Glycogen storage disorders.
P3, lines 101-105. GSDs are not caused by fasting hypoglycaemia, etc but GSDs lead to hypoglycaemia, hepatomegaly etc.
A figure showing the cycle of glycogen metabolism should be presented identifying the key enzymes involved in GSDs.
Mitochondrial metabolic dysfunction.
This section can be reduced considerably. There is no need to describe the basic electron transport chain. It should be pointed out that reactive oxygen species (ROS) are produced during normal mitochondrial respiration but the level is obviously increased in GSDs. The mechanism that underlies this alteration in GSDs should have been explored further.
Lines 127-129. The lactate-pyruvate interconversion does not occur due to glycogen metabolism per se but occurs in response to energy requirements allowing a temporary switch to anaerobic respiration. Presence of GSDs may increase the dependence on lactate formation leading to problems of mitochondrial function.
Lines 130-138: This is irrelevant as the review is concerned with the mutations in the enzymatic machinery of glycogen metabolism and should be removed.
Lines 138-149: Would you expect these changes in enzyme activity if glycogen hydrolysis was impaired? Are the enzymatic responses due to inadequate supply of glycogen derived glucose?
Overproduction of ROS and oxidative stress.
This section is redundant as the increased ROS levels in GSDs have been addressed in previous sections.
Lines 270-272: In GSDs, how would increasing mitochondrial biogenesis reduce oxidative stress given the supply of glucose remains restricted?
Author Response
Reviewer 2 comments
Dear Reviewer,
We sincerely appreciate your thorough review of our manuscript and the valuable comments you provided. In response, we have addressed all your queries with point-by-point clarifications (in bold letters). All the modifications have been highlighted in yellow within the main text of the manuscript.
General comments:
- The paragraphs are extremely long and therefore difficult to follow. Shorter paragraphs with a logical progression are advised. It is well known that glycogen storage disorders are associated with mitochondrial dysfunction. This review should have explore how reduced glycogen synthesis or abnormal degradation impacts on the energy provision pathways in the cell and suggest how this can lead to functional abnormalities in the mitochondrion.
We tried to shorten the information and focused throughout the manuscript on abnormal glycogen accumulation and glycogen metabolism which leads to mitochondrial dysfunction.
Specific comments:
Glycogen storage disorders.
- P3, lines 101-105. GSDs are not caused by fasting hypoglycaemia, etc but GSDs lead to hypoglycaemia, hepatomegaly etc.
Corrected
- A figure showing the cycle of glycogen metabolism should be presented identifying the key enzymes involved in GSDs.
Figure 1 shows the deficiency of key metabolic enzymes implicated in GSDs .
- Mitochondrial metabolic dysfunction.
This section can be reduced considerably. There is no need to describe the basic electron transport chain. It should be pointed out that reactive oxygen species (ROS) are produced during normal mitochondrial respiration but the level is obviously increased in GSDs. The mechanism that underlies this alteration in GSDs should have been explored further.
We have revised the section. Basic ETC information was removed. Information on ROS production is added. This section consists of preliminary information on factors responsible for mitochondrial dysfunction. The detailed mechanism has been explained under individual subheadings.
Lines 127-129. The lactate-pyruvate interconversion does not occur due to glycogen metabolism per se but occurs in response to energy requirements allowing a temporary switch to anaerobic respiration. Presence of GSDs may increase the dependence on lactate formation leading to problems of mitochondrial function.
Agree and modified as suggested
- Lines 130-138: This is irrelevant as the review is concerned with the mutations in the enzymatic machinery of glycogen metabolism and should be removed.
Agreed. Removed.
- Lines 138-149: Would you expect these changes in enzyme activity if glycogen hydrolysis was impaired? Are the enzymatic responses due to inadequate supply of glycogen derived glucose?
We mentioned and discussed study findings related to mitochondrial dysfunction in GSDs, as reported by Kurbatova et al. 2014 (53 in the reference list). This study highlighted that enzymatic activities were impacted across various GSDs. Since GSDs primarily arise from defects in enzyme activities during glycogen metabolism, the observed effects on enzymatic function or delays in response could be attributed to inadequate glycogenolysis. However, further research is necessary to thoroughly investigate and address these mechanisms.
- Overproduction of ROS and oxidative stress.
This section is redundant as the increased ROS levels in GSDs have been addressed in previous sections.
In the previous section, we briefly discussed the factors that may contribute to mitochondrial dysfunction in relation to GSDs. In the following section, we dwel deeper into the mechanisms underlying this dysfunction and illustrate the process with a diagram (Figure 3).
- Lines 270-272: In GSDs, how would increasing mitochondrial biogenesis reduce oxidative stress given the supply of glucose remains restricted?
- We mentioned that increased mitochondrial biogenesis reduces ROS production and oxidative stress, leading to improved mitochondrial function. This concept is supported by the following study (Carling et al. Cell metabolism. 2015;21(6):799-804) - 83 in the reference list.
Thank you
Best Regards
Kumudesh

Reviewer 3 Report
Comments and Suggestions for Authors
Dear authors,
Please see below my comments
1) Table 1 is very big (3 pages), needs to be smaller because interrupts the overall flow of manuscript or table to be included at the end.
2) Figure 1, not aligned correctly causing issues with the text.
3) Lack of figures on the other sections.
4) ROS-Should include in detail how genes and proteins are modified. The mechanism that causes toxicity and cancer is important and currently missing from manuscript.
Comments on the Quality of English Language
Dear authors,
please check again the structure of sentences
Author Response
Reviewer 3 comments
Dear Reviewer,
We sincerely appreciate your thorough review of our manuscript and the valuable comments you provided. In response, we have addressed all your queries with point-by-point clarifications (in bold letters). All the modifications have been highlighted in yellow within the main text of the manuscript.
- Table 1 is very big (3 pages), needs to be smaller because interrupts the overall flow of manuscript or table to be included at the end.
We have reduced the size of Table 1 by decreasing the font size to enhance reader convenience
- Figure 1, not aligned correctly causing issues with the text.
The alignment in Figure 1, now referred to as Figure 2, has been adjusted accordingly
- Lack of figures on the other sections.
We have also included Figure 1, Figure 3, Figure 4, and an additional
Table 2 to further elucidate the mechanism
- ROS-should include in detail how genes and proteins are modified. The mechanism that causes toxicity and cancer is important and currently missing from manuscript.
We discussed the mechanism of ROS production, oxidative stress, and its scavenging, which presented in Figure 3 . While this review focuses on GSD, but we highlighted the ROS mechanism and cancer which illustrated in Figure 3
Thank you
Best Regards
Kumudesh

Round 2
Reviewer 2 Report
Comments and Suggestions for Authors
The manuscript has been improved considerably.
Author Response

(The authors gave the same response as above.)
